# Defining, identifying and addressing problematic polypharmacy within multimorbidity in primary care: a scoping review

Jung Yin Tsang ![ORCID] ,[1,2] Matthew Sperrin ![ORCID] ,[2,3] Thomas Blakeman,[1,2] Rupert A Payne,[4] Darren Ashcroft[2,5]

For numbered affiliations see end of article.

**Correspondence to**
Dr Jung Yin Tsang;
jungyin.tsang@manchester.ac.uk

## ABSTRACT

**Introduction** Polypharmacy and multimorbidity pose escalating challenges. Despite numerous attempts, interventions have yet to show consistent improvements in health outcomes. A key factor may be varied approaches to targeting patients for intervention.

**Objectives** To explore how patients are targeted for intervention by examining the literature with respect to: understanding how polypharmacy is defined; identifying problematic polypharmacy in practice; and addressing problematic polypharmacy through interventions.

**Design** We performed a scoping review as defined by the Joanna Briggs Institute.

**Setting** The focus was on primary care settings.

**Data sources** Medline, Embase, Cumulative Index to Nursing and Allied Health Literature and Cochrane along with ClinicalTrials.gov, Science.gov and WorldCat.org were searched from January 2004 to February 2024.

**Eligibility criteria** We included all articles that had a focus on problematic polypharmacy in multimorbidity and primary care, incorporating multiple types of evidence, such as reviews, quantitative trials, qualitative studies and policy documents. Articles focussing on a single index disease or not written in English were excluded.

**Extraction and analysis** We performed a narrative synthesis, comparing themes and findings across the collective evidence to draw contextualised insights and conclusions.

**Results** In total, 157 articles were included. Case-finding methods often rely on basic medication counts (often five or more) without considering medical history or whether individual medications are clinically appropriate. Other approaches highlight specific drug indicators and interactions as potentially inappropriate prescribing, failing to capture a proportion of patients not fitting criteria. Different potentially inappropriate prescribing criteria also show significant inconsistencies in determining the appropriateness of medications, often neglecting to consider multimorbidity and underprescribing. This may hinder the identification of the precise population requiring intervention.

**Conclusions** Improved strategies are needed to target patients with polypharmacy, which should consider patient perspectives, individual factors and clinical appropriateness. The development of a cross-cutting measure of problematic polypharmacy that consistently

---

## STRENGTHS AND LIMITATIONS OF THIS STUDY

⇒ This is the first scoping review to explore and conceptualise how patients with problematic polypharmacy are targeted for intervention
⇒ It includes multiple types of evidence, including systematic reviews, quantitative, qualitative and mixed methods studies, along with policy documents.
⇒ Our synthesis capitalises on the shared challenges involved in managing both polypharmacy and multimorbidity with a greater focus on articles regarding polypharmacy in chronic conditions rather than acute medication adjustments.
⇒ It was not always possible to separate results in studies encompassing both primary and secondary care.

---

incorporates adjustment for multimorbidity may be a valuable next step to address frequent confounding.

## INTRODUCTION

Polypharmacy in multimorbidity is an increasing global priority.[1] With an ageing population, over a quarter of the population are living with multiple long-term conditions also known as multimorbidity.[1] This is often associated with polypharmacy, which is broadly defined as the use of multiple medications.[2] Medications carry clear benefits, yet the use of multiple medicines can be linked to adverse consequences, including increased treatment burden, unplanned hospitalisation and death.[3 4] For single conditions, people with more severe disease often require more medications. For example, the National Institute for Health and Care Excellence (NICE) guidelines recommend six medicines to be initiated post myocardial infarction for secondary prevention.[5] Yet in multimorbidity, the number of medicines quickly add up, with limited evidence of benefit over risk as this population is frequently excluded in trials.[6] As the number of medicines prescribed

increases, so does the direct risk of adverse drug reactions, increasing health service costs and utilisation, reducing adherence and decreasing quality of life.[7-9] This can be particularly problematic for older patients, for whom prescribing is more common and thus associated with greater possibility of prescribing errors. Moreover, the risks of harms are increased due to changes in pharmacokinetics (eg, impaired drug metabolism, changes in drug binding) and pharmacodynamics (eg, increased sensitivity to adverse effects).[10-12] Problematic polypharmacy has previously been defined as 'the prescribing of multiple medications inappropriately, or where the intended benefit of the medication is not realised'.[3]

Despite numerous interventions targeting polypharmacy, there remains little evidence of improvement of health outcomes, such as hospitalisations and death.[13-15] However, some reductions in inappropriate prescribing have been observed. Successes of these interventions have been highly variable and greatly affected by differences in implementation and targeting of patients.[13-15] Further conceptualising the complex and varied approaches to targeting patients with problematic polypharmacy and multimorbidity may inform empirical research and improve future intervention design.[2] Therefore, a scoping review was performed, to adopt an effective approach for assessing a broad evidence base. This review centres on considering the pivotal role of primary care professionals and capitalises on the shared challenges involved in managing polypharmacy and multimorbidity. The overarching aim of the review was to explore how patients are targeted for intervention by examining the literature with respect to (1) understanding how polypharmacy is defined; (2) identifying problematic polypharmacy in practice; and (3) addressing problematic polypharmacy through interventions.

## METHODS

A scoping review as defined by the Joanna Briggs Institute was performed consistent with the Preferred Reporting Items for Systematic Reviews extension for Scoping Reviews (PRISMA-ScR) guidance.[16] This allowed an exploration of both breadth and depth of the topic, which was imperative given the complexity and heterogeneity of evidence. We purposely retained multiple types of evidence (eg, randomised controlled trials (RCT), consensus trials and qualitative video ethnography) to allow learning through quantitative, qualitative and mixed methods studies, as well as policy and grey literature, to increase relevance and examine the latest evidence base to date.

### Search strategy

A literature search was conducted within Medline, Embase, Cumulative Index to Nursing and Allied Health Literature and Cochrane Database of Systematic Reviews in January 2023. Search terms were developed after a preliminary search of articles covering our population, concept and context of interest, provided in table 1.

**Table 1** Search terms used

| Category | Search terms used |
|---|---|
| Population: *multimorbidity* | Multimorbid* or multiple long-term conditions or multiple health conditions |
| Concept: *problematic polypharmacy* | Polypharmacy or polypharmacotherapy or hyperpolypharmacy or polymedicine* or polimedicin* or multiple medic* or multimedic* or inappropriate prescrib* or overprescrib* or underprescrib* or deprescrib* |
| Context: *primary care* | Primary care or primary healthcare or general practi* |

This included the population of people with multimorbidity, the concept of problematic polypharmacy and the context of primary care. We limited our final search strategy to include only articles from 2004 onwards based on the earliest date of relevant articles from a preliminary search. Three additional databases were then searched for grey literature and clinical trial records: ClinicalTrials.gov, Science.gov and WorldCat in February 2023. We then followed an iterative process of snowballing through a supplementary search of references, citation lists and related articles using Google Scholar. Consistent with scoping reviews guidance, critical appraisal was not undertaken. An updated search was then completed in February 2024.

### Eligibility criteria

The eligibility criteria with typical exclusion examples are presented in table 2, guided by the Population, Concept and Context framework recommended by PRISMA-ScR[16]:

### Study selection

Studies meeting the inclusion criteria were initially selected, based on screening the titles, abstracts and subsequent full papers by one researcher (JT). A random selection of 10% the records was analysed independently by a second researcher (TB) with 97% agreement of inclusion. Disagreements were resolved through discussion with the wider team.

### Data extraction and analysis

The data were extracted from eligible studies using a standardised data extraction form and included the author, year of publication, country of origin, type of the publication, polypharmacy definitions, type of participants, descriptions of interventions (if applicable) and key findings (see additional file 1). Further elaboration of the extracted data involved grouping studies according to their focus on either defining, identifying and addressing polypharmacy, with some spanning multiple elements. The main analysis took the form of a narrative synthesis, using mainly qualitative descriptive data consistent with PRISMA-ScR guidance.[16] This compared themes and findings from grouped studies and using the collective evidence to draw contextualised insights and conclusions.

**Table 2**  Eligibility criteria and typical exclusion examples

| Inclusion criteria | Typical exclusion examples |
|---|---|
| Population — adults living with multimorbidity: <br>▶ Studies must include adults (18 years and older) <br>▶ Studies must focus on those with multimorbidity—defined as 2 or more long-term conditions, not linked to an 'index disease' | ▶ Studies focusing on patients with diabetes with renovascular disease (ie, has an index condition of diabetes) |
| Concept — problematic polypharmacy: <br>▶ Studies focusing on polypharmacy—defined as the concurrent use of multiple medications <br>▶ Studies that consider the long-term clinical impact of multiple medicines <br>▶ Studies that consider the consequences of multiple medicines or the 'problematic' element of polypharmacy | ▶ Studies focused on single medications <br>▶ Studies based on prescribing of antibiotics for acute presentations only <br>▶ Studies that are simply descriptive of the number of tablets taken and do not report any risk factors, outcomes or consequences |
| Context — primary care: <br>▶ Studies with relevance to primary care, including studies which crossed the primary-secondary care interface. | ▶ Studies solely on hospital-based pharmacists |
| Study type <br>▶ Studies written in English <br>▶ Studies presenting full descriptions of the research (eg, research studies, systematic reviews, randomised controlled trials, pilot studies and policy documents) | ▶ Letters, comments, conference abstracts, protocols, proceedings and so on. |

## RESULTS

The search yielded 727 unique articles, with the process illustrated in figure 1. During eligibility screening, 486 were excluded after assessment of the abstract and 84 further full-text articles were excluded. A total of 157 articles were included in the final synthesis (online supplemental file 1), of which 19 were added during the updated search. This included 52 meta-analyses and reviews, 55 quantitative (including 9 RCTs and 19 longitudinal analyses), 36 qualitative studies (including 6 consensus studies and 2 RCT evaluations), 9 pilot or feasibility studies and 5 policy documents. The literature was

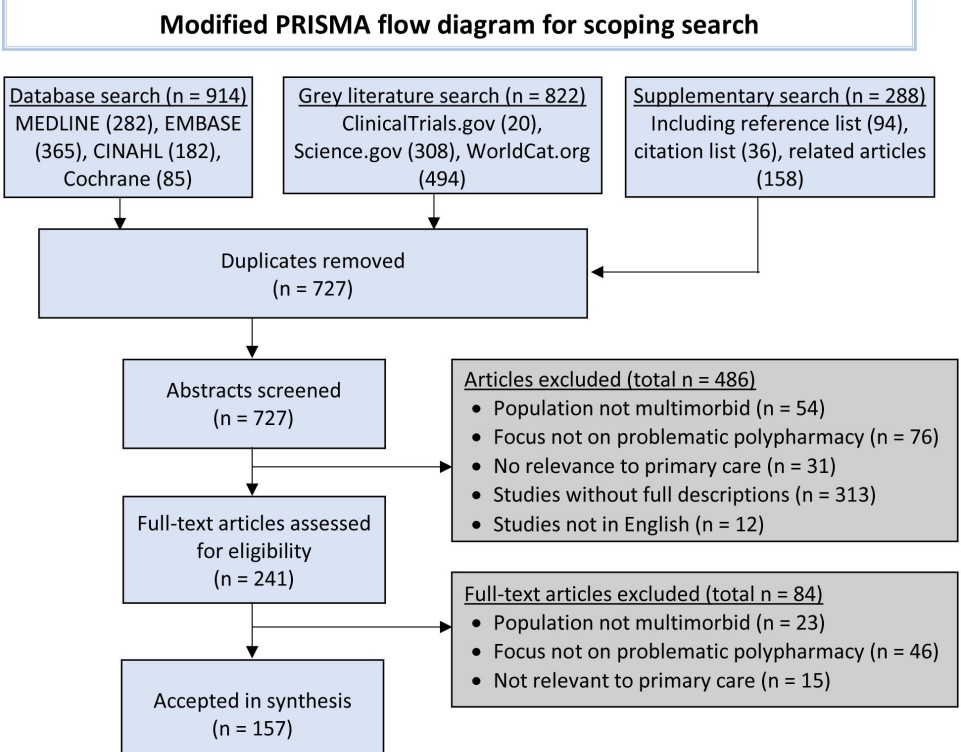

**Figure 1**  A Preferred Reporting Items for Systematic Reviews and Meta-Analyses flow diagram illustrating search results.

**Table 3** Illustrative list of examples for polypharmacy definitions

| Definitions | Descriptions/examples |
|---|---|
| Quantitative definitions | |
| Single cut-offs of medication count | ≥2, ≥3, ≥4, ≥5, ≥8, ≥10, ≥11 or ≥20 medications |
| Single cut-offs of a medication group | >2 anticholinergic medications<br>>3 antipsychotic medications |
| Groups of medication counts | 0–4 medications, 5–9 medications, 10–14 medications, ≥15 medications<br>0–5 medications, 6–8 medications, 9–11 medications, ≥12 medications<br>0–6 medications, 7–9 medications, 10–13 medications, ≥14 medications |
| Categorisation with levels or attributes | Mild polypharmacy 1–4 or 2–3 medications<br>Minor polypharmacy 2–4 medications<br>Major polypharmacy ≥5 medications<br>Standard polypharmacy 5–9 or 6–9 medications<br>Severe polypharmacy ≥6 or ≥10 medications<br>Extreme polypharmacy ≥10 medications<br>Hyperpolypharmacy ≥10 medications<br>High-level polypharmacy ≥10 medications |
| Qualitative definitions | |
| Overprescribing | More medications than clinically indicated or unnecessary medications or presence of medications with no clinical indications or for which a safer alternative exists |
| Underprescribing | Lack of an indicated medication, or prescribed an inadequate amount or prescribed less frequent than appropriate |
| Drug-drug interactions | Any potential interaction, or harmful combination |
| Inappropriate medications | Defined by set criteria, for example, overprescribing, misprescribing and potential interactions |
| Prescribing cascade | Medication prescribed to treat the side effect of another medication |
| Absence of indication | Medication not matching the diagnosis |
| Therapeutic duplication | Same medicine used more than once or twice within the same therapeutic group used (eg, multiple antidepressants) |
| No therapeutic benefit | Medications with lack of effectiveness |
| Not cost-effective | Availability of an equally effective, lower cost alternative |

Illustrative examples of wide range of definitions for polypharmacy used in the literature.[18 19] Generally, quantitative definitions focus on operationalising medication count, regardless of whether polypharmacy is problematic whereas most qualitative definitions attach descriptors to describe scenarios where polypharmacy may be clinically problematic.

varied with international articles covering a range of polypharmacy issues, from definitions to interventions, with some focussing on subpopulations with multimorbidity (eg, frailty) and subcategories within the broader context of primary care (eg, residential care facilities).

### Understanding how polypharmacy is defined
#### Numerous polypharmacy definitions
There is no consensus on a definition for polypharmacy, with significant variations in approaches to targeting problematic polypharmacy.[2 3 17] Over 100 definitions of polypharmacy have been used, reflecting the discordance of approaches.[18 19] Two main approaches to defining polypharmacy can be grouped into quantitative (using a form of medication count) and qualitative definitions (using descriptive notions of prescribing quality), with some studies using a combination of these definitions. Table 3 gives illustrative examples of these definitions.

Quantitative definitions of polypharmacy were more frequent, with over 90% of publications using some form of medication count.[2 18–21] For example, the WHO defines polypharmacy as four or more medicines, academic studies most commonly use 5 or more.[1 2] Other quantitative definitions included categorisations rather than cut offs of medication count. These were frequently labelled levels (eg, mild, moderate and severe) or attributes (eg, excessive, extreme), yet counts within these categories were also inconsistent.[12 18 19 22–24] Generally, quantitative definitions were easier to operationalise and more reproducible, with a focus on medication count, regardless of whether polypharmacy is problematic. In contrast, qualitative definitions largely required clinical judgement to evaluate prescribing quality, carrying a focus on when polypharmacy becomes problematic. This frequently highlighted the overuse or overprescribing of medications. But definitions also covered aspects of misprescribing, often through applying a list of defined prescribing criteria, and also underprescribing, though only a few studies emphasised this aspect. The terms

'appropriate', 'inappropriate' and 'problematic' polypharmacy were also commonly used to describe when multiple medications were justified compared with when the clinical indication was unclear.[3 18 19 25] These definitions have now been expanded to cover further dimensions of polypharmacy, such as the increasing recognition of the importance of patient and carer input in determining the appropriateness of medications.[26 27]

### The challenges of defining when polypharmacy is 'problematic'

The understanding of polypharmacy has progressed over time, with an increasing shift to more clinically applicable definitions. This reflects the increasing complexity of decision-making for combinations of medicines tailored to individual needs. There is also recognition that it is not possible to account for clinical appropriateness through simple medication counts.[18 19] Commonly people with multiple health needs may well be appropriately prescribed more than 10 medications for therapeutic and symptomatic benefit, which would be termed extreme polypharmacy in some studies and guidelines.[28 29] Yet there is some validity to numeric approaches as increasing medications are strongly associated with drug-related problems, and very high counts of medication are usually questionable.[30] There is also a need to improve the consistency of reporting medication exposure characteristics.[18 19 31–33] Various definitions have been used to define temporality and 'long-term' use, with some publications including 'acute' and 'as required' medications as opposed to chronic medications, with varied definitions of time periods (ranging from 1 to 240 days).[18 19] Terms such as problematic polypharmacy and inappropriate polypharmacy have been increasingly favoured, as they consider appropriateness and clinical decision-making.[34] Yet qualitative research suggests that these labels were still insufficient to reflect the complexity of medicines management, with practitioners juggling terms such as 'potentially inappropriate' and 'specifically appropriate' and others considering them 'judgemental' and even 'accusatory'.[35]

### Identifying problematic polypharmacy in practice
#### Targeting potentially high-risk populations

Various strategies target higher risk populations to try and identify problematic polypharmacy. One common approach uses simple cut offs of age (commonly ≥65 years) combined with cut offs of medications (frequently ≥5) and this was the main inclusion criterion for the majority of trials.[13] Another approach adopted by multiple national recommendations advocate case finding through high-risk groups.[36–38] For instance, both NICE guidelines and the Australian Commission on Safety and Quality in Healthcare recommend greater attention for older people with frailty, and complex multimorbidity and co-existing mental and physical health problems.[2 29 36] Accordingly, several national indicators, initiatives and studies also use combinations of these approaches.[36–43] Other approaches include risk scores to identify patients at high risk of particular outcomes (eg, hospitalisations or adverse drug reactions) but these require further development.[44 45] Overall, strategies to identify potentially high-risk populations currently demonstrate variable validity in polypharmacy and are seldom comprehensive or holistic, as they are specific to the needs of particular groups.[2 36 46]

#### Targeting potentially inappropriate medicines

Evaluating the appropriateness of individual medications is a common approach both as a case-finding approach and as a surrogate measure of prescribing quality across polypharmacy. Various tools have been developed to identify potentially inappropriate medicines and these can be split into explicit and implicit tools, with some tools combining both (examples in table 4).[47–49] The majority have been developed using expert opinion and consensus methodology, and originally were designed for evaluating individual medications, rather than polypharmacy as a whole.[47 50] Explicit tools contain specific criteria or scenarios leading to potential adverse drug events and carry advantages of reproducibility and ease of automation.[51–55] Implicit tools require judgement, which means they can be subjective and demand more time and clinical expertise. Nevertheless, explicit tools are limited to specific drugs and diseases, but implicit tools can be applied to any medication. This perhaps allows implicit tools greater applicability in polypharmacy, as explicit tools will miss out any medicines outside criteria.[56]

Several systematic reviews have revealed a high level of variability of included criteria within explicit tools.[47–50 54 57] A review of 36 explicit tools reported criteria spanning 907 medications and medication classes, but only 44 medications and 4 classes were reported by the majority.[48] This was despite over 85% of these tools being developed based on either the Beers or the Screening Tool of Older Person's Prescriptions/Screening Tool to Alert doctors to Right Treatment (STOPP/START) criteria.[48] Due to this, many studies combine several explicit criteria to complement the list of medications included.[47 48 50 58–62] Only about a third of tools suggested alternative treatments to potentially inappropriate medicines, yet nearly 70% of suggested alternatives were deemed inappropriate by other tools.[47] Implicit tools are also diverse in nature, with reviews identifying over 16 different tools incorporating implicit criteria.[54 63] These ranged from risk scores to lists of questions specifying appropriate use or criteria to evaluate the administrative burden to patients.[54 63–65] Several tools combine implicit and explicit indicators, including documents used for national guidance (eg, Australian Prescribing Indicators Tool).[63 66 67]

### Key limitations in identifying problematic polypharmacy in practice

Current strategies to identify problematic polypharmacy demonstrate inadequate performance. At present, risk stratification tools remain too broad, and seldom consider the clinical appropriateness of individual medications.[34 68] Though comprehensive explicit criteria are helpful in identifying potentially inappropriate

**Table 4** Key examples of explicit and implicit tools of appropriate prescribing

| Tool | Description | Strengths | Limitations |
|---|---|---|---|
| Beers criteria (*Explicit tool*) | ▶ First widely used explicit criteria<br>▶ Contains over 200 criteria (2023 version) including potentially inappropriate medications to be avoided such as drug disease and drug–drug interactions, particularly in older adults. | ▶ International studies have shown predictive validity for adverse drug reactions, falls, cognitive function, hospitalisation and death.<br>▶ Endorsed by the American Geriatric Society and updated approximately every 3–4 years.<br>▶ Easier to automate in drug records as criteria are specific | ▶ No positive clinical outcomes in RCTs to date<br>▶ No prioritisation of medications for review<br>▶ Can be challenging to use as long list of criteria<br>▶ Does not address underprescribing<br>▶ Focus is on individual medications rather than polypharmacy as a whole |
| Screening Tool of Older Person's Prescriptions/ Screening Tool to Alert doctors to Right Treatment—STOPP/ START (*Explicit tool, but newer versions also contain implicit measures*) | ▶ One of the most widely used explicit criteria globally for older adults<br>▶ Contains 133 criteria for potentially inappropriate medications, and 57 potential underprescribing criteria (version 3), organised according to medication and disease groups | ▶ Some positive outcomes shown in several RCTs<br>▶ Also addresses aspects of underprescribing in addition to overprescribing<br>▶ Easier to automate in computerised drug records as most criteria are specific | ▶ Misses out medications out of criteria<br>▶ Can be challenging to use as long list of criteria<br>▶ No prioritisation of medications for review<br>▶ Focus is on individual medications rather than polypharmacy as a whole |
| Medication Appropriateness Index— MAI (*Implicit tool*) | ▶ First widely used implicit criteria<br>▶ Lists 10 criteria that evaluate various aspects of medication appropriateness (eg, indication, effectiveness, dose) | ▶ Some positive outcomes shown in several RCTs<br>▶ Can be applied to all medicines | ▶ Time consuming to execute<br>▶ Requires clinical expertise and can be subjective<br>▶ Difficult to automate<br>▶ No prioritisation of medications for review<br>▶ Focus is seldom on polypharmacy as a whole or underprescribing |
| Drug Burden Index—DBI (*Implicit tool, as requires further judgement to evaluate appropriateness after calculating score*) | ▶ Widely researched risk score<br>▶ Calculates the cumulative exposure of sedatives and anticholinergics to give a score between 0 and 1. | ▶ International studies have shown predictive validity for falls, fractures, general practice visits and admission.<br>▶ Takes into account licenced doses to allow transferability between counties<br>▶ Easier to automate in drug records. | ▶ No positive clinical outcomes in RCTs to date<br>▶ No consideration for appropriateness or specific indication of medicines<br>▶ Only focused on sedatives, and anticholinergics<br>▶ Can be challenging to calculate at point of care unless computerised<br>▶ Does not address polypharmacy as a whole or underprescribing |

A descriptive summary of selected examples of widely studied explicit and implicit tools.[48 54 174–177]
RCT, randomised controlled trial.

medications, translation into everyday care remains elusive due to challenges in clinical application, and the omission of medications not included in criteria.[48 69] For instance, previous studies have found that less than 25% of adverse drug reactions are caused by drugs listed by Beers criteria.[70 71] Additionally, apart from STOPP/ START, most widely used tools were not designed to also cover underprescribing (table 4), with some studies also choosing to omit many of the underprescribing criteria in its application.[47–50 54] Furthermore, there have been questions as to the utility of long lists of medications as studies have shown a high prevalence of potentially inappropriate medications (over 30% of patients) but low variability within many criteria, potentially leaving little room for improvement.[72] Studies also mention usability issues with such long lists, even with computerised integration, and the difficulties of making treatment decisions without prioritisation of criteria, particularly as their predictive validity is unknown.[47 59 68 73 74] Still, as the majority of instruments were developed focussing on patients over 65 years

old, the suitability for middle-aged adults is unknown, yet both polypharmacy and multimorbidity are increasing in this age group.[20 30 75] Only a handful of criteria have been developed and validated (eg, Prescribing Optimally in Middle-aged People's Treatments criteria), all including significantly fewer criteria for individual medications and medication classes.[54 56 63 76 77] Again, this further limits applicability in problematic polypharmacy, where the whole of the medication regimen should be considered.

### Addressing problematic polypharmacy through interventions
#### Large variability in interventions addressing polypharmacy
Interventions to address problematic polypharmacy have covered a wide range of aims, such as reducing adverse drug reactions, increasing the appropriateness of medicines use, reducing falls, improving patient adherence and maintaining quality of life.[13 78–81] To combat overprescribing specifically, deprescribing interventions have also received significant attention, though interventions that focus on underprescribing are much less.[82–86] Several

large reviews highlight good evidence of improving prescribing patterns, yet mixed and low certainty of evidence in improving patient-relevant outcome measures.[2 13–15 80 87–90] Reviews covering over 150 primary studies reported no differences in all-cause mortality and no clear evidence of benefit in reduced hospitalisations, when comparing interventions to usual care.[13 80 88 91–94] There were also no differences in quality of life, adverse drug reactions, readmission rates, primary care visits and emergency department visits.[13 80 92–94] Two reviews have highlighted some economic benefits in reducing healthcare expenditure, but others highlight inconsistencies due to low-quality evidence.[92 93 95] Overall, there is evidence that these interventions are safe and do not lead to harm, but may still be time and resource intensive for both patients and practitioners, as many require continuing input.[13 80 82] Likewise, mixed evidence of improved clinical outcomes, such as falls, is also observed even in more focused populations, such as those with frailty and in long-term care facilities.[84–86 96]

### Multiple intervention components to address polypharmacy, with unclear effectiveness

The majority of polypharmacy interventions were multimodal with a review revealing 14 different elements from 80 studies and an average of 2.5 elements per intervention.[13 97] The most common elements included medication reviews, training for professionals and using tools, such as clinical decision support, checklists or audit and feedback.[13 43 74 97–100] Other components strengthened interprofessional or multidisciplinary collaborations by involving clinical pharmacists, nurses or geriatricians.[13 92 94 97 100–107] There were also patient-facing components, such as education and training for patients and patient interviews to seek their understanding and lived experiences with their medicines.[108–113] Despite the growing literature on the importance of patient-centred care in medicines management, current literature highlights that patient priorities are seldom fully integrated into polypharmacy interventions.[13 82 91 97 114–121] Patient-centred approaches also appear to be key to improving adherence, as a frequent discordance between practitioner and patient views is reported.[13 15 97 122–128] More recent interventions that do adopt a patient-centred model show some mixed improvements in appropriate prescribing, but limited improvements in outcomes, reflecting some of the challenges of integrating patient priorities into routine medication reviews.[99 108–113 129 130]

In terms of effectiveness of individual intervention elements, similar effect sizes have been observed in reducing the number of potentially inappropriate medicines, with no particular components showing particular superiority.[13 80 97] However, generalised professional education programmes appear to be less effective than individualised interventions.[13 131] Medication reviews are also the most commonly adopted component, but as a single intervention, there remains insufficient evidence of medication reviews alone improve clinical outcomes.[84 132 133] Despite the advantages of automation, electronic tools in trials demonstrate high variability in implementation within large pan-European and global trials, and no clear positive advantages on relevant patient outcomes have been reported.[13 134 135] Pharmacists show promise as an extra resource for managing polypharmacy in individual studies, but two recent reviews revealed uncertain effects on optimising medicines.[92 94 102–106 136 137] Community pharmacists can contribute to medication safety, but more in-depth management such as polypharmacy medication reviews was seen as outside the scope of community pharmacy.[105 138 139]

### Key challenges in addressing problematic polypharmacy

In spite of the breadth of interventions targeting polypharmacy, it remains unclear which intervention components are more important.[13] Theory-informed interventions are few and there are opportunities for improvements in intervention design through stronger foundations on theoretical frameworks and behaviour change techniques.[128 140–144] Widespread variation exists in the everyday management of medicines and polypharmacy.[2 3 145–147] These variations occur at patient, prescriber, regional and international levels, and indicate links between problematic polypharmacy and health inequalities.[1–3 39 145 146 148–150] As such, multiple challenges to addressing problematic polypharmacy need to be overcome, going beyond the identification of individual barriers and facilitators and translating these into practice within the complexity of interlinked systems of care.[2 39 151 152] The failure of the implementation of interventions is commonly down to the lack of consideration of integration into an already high-demand system in everyday primary care.[152–155]

For patients with polypharmacy and multimorbidity, prioritisation and decision-making are a challenge, given that they can receive 10 times the amount of information during consultations due to compounding health issues, interacting medications and complex social issues.[156] Yet patient priorities and shared decision-making are vital to deciding the appropriateness of medications, so improvements need to be made to both the clarity of information provided and the integration of patient views into polypharmacy decisions.[2 26 27 114 118 121 128 130 156 157] The majority of patients appear willing to discuss deprescribing medications, particularly if they have a good relationship with their doctor.[82 105 118 135 155] However, they also have strong beliefs and attitudes of the value of their medicines, with inertia generated when feeling well on their current medication regimen.[82 118 120 152 158–160]

For health professionals, work and effort are required to even consider deprescribing, particularly as prescribing is so embedded in routine practice and finding an appropriate time to initiate the discussion is often difficult given competing priorities.[153 154 161–163] A comprehensive polypharmacy medication review is described as 'impossible' to complete in 10 minutes, leading to practitioners defaulting to a swifter review and degrading

medication reviews to being 'mundane' tasks.[158] This is combined with the work to gain awareness (of new policies, guidelines and tools), overcome significant uncertainty in evidence (with 'unmeasurable' risk-benefit) and increase self-efficacy with limited resources and alternatives.[149 154 162–167] On an organisational and systems level, fragmentation of care and poor coordination between healthcare teams and specialists often lead to deferring ownership of deprescribing, and miscommunication to patients, leading to medication-related problems.[149 151 161 166 168 169] More comprehensive approaches and better resources are needed to support practitioners and organisations in pushing for improved polypharmacy decisions in a patient-centred manner, rather than simply maintaining the 'status quo'.[35 82 148 162 164]

## DISCUSSION

The evidence highlights significant challenges to optimising the targeting of patients with problematic polypharmacy for intervention. Despite the extensive number of studies, there is little evidence of improved patient outcomes even for higher risk populations, including individuals with frailty and those in long-term care facilities. This is highly suggestive that the targeting of patients with problematic polypharmacy needs to be more focused or even that the incorrect populations and medications are currently being targeted. Simple counts or 'at-risk' populations appear too broad as case-finding approaches. Though potentially inappropriate prescribing criteria can be helpful, this approach is also inadequate as it omits many patients not fitting criteria, lacks consistency across criteria and often overlooks underprescribing and multimorbidity. Furthermore, given the complexity of prescribing decisions in multimorbidity and the importance of considering patient values, potentially inappropriate criteria can rarely be used alone in assessing appropriateness. Due to the frequent confounding of multimorbidity observed in studies evaluating polypharmacy outcomes, coupled with the diverse combinations of medications involved in adverse drug reactions, there is a need for cross-cutting tools that can effectively capture the interplay of multiple health conditions in patients.[91 147] Ultimately, the targeting of patients with problematic polypharmacy need to take into account patient perspectives, individual factors and clinical appropriateness.

### Implications for further research and practice

The approach to targeting patients needs to be improved as a first step, which may allow the identification of an optimal population for polypharmacy interventions. A next step to enhance clinical utility may be the routine adjustment of multimorbidity, as there is frequent confounding of polypharmacy outcomes within studies.[91] In doing so, we may be able to identify patients who are both overprescribed and underprescribed medicines yet consider some degree of clinical appropriateness. An opportunity exists to produce a cross-cutting measure

beyond single diseases and individual drug interactions to evaluate patients as a whole, with the aim of improving overall health.[68 164]

The multifactorial drivers of polypharmacy also mean that approaches to address problematic polypharmacy need to go beyond targeting patients and practitioners alone.[39 152] Despite this, evidence of a systems approach encompassing policy-makers, organisations, practitioners, patients and carers is lacking.[2 39 151 152] Both the growth of evidence-based medicine and desire to minimise all risk are significant drivers of increased medicines burden and problematic polypharmacy. Yet polypharmacy is rarely 'evidence-based', as it would be impossible to have a large enough sample size to perform drug trials and meta-analyses of the millions of combinations that patients with multimorbidity are taking.[6 170] Studies examining exclusion criteria of RCTs estimate that over 90% of this population would be excluded from trials, questioning their representativeness.[171] The emphasis on following guidelines and increasing treatment intensity should be balanced with the understanding that high-quality personalised healthcare can only be achieved through also carefully reducing, stopping or not initiating medication, with shared decision-making and agreed care objectives.[172 173]

### Strengths and limitations

This scoping review syntheses a wide breath of literature to explore the existing evidence. It allowed a systematic approach on an initial search strategy and was also adaptable to heterogeneous sources (eg, policy documents) and developing literature (eg, pilot studies) through related article, supplementary and grey literature searching. It examined the overlapping concepts of polypharmacy and multimorbidity concurrently, allowing synergies in evidence generation and critique.

There are several limitations of our review to consider. As with other scoping reviews, critical appraisal was not performed. Polypharmacy is an area that has received widespread attention, with hundreds of primary studies and dozens of systematic reviews. Hence, in our attempts to present generalisable findings, the nuances within primary studies may be lost, such as differences in study setting, population or intervention characteristics. While we made efforts to specifically extract primary care-related findings, it was not always possible to separate results in studies encompassing both primary and secondary care. Furthermore, by emphasising multimorbidity and primary care in our search, we may have overlooked research investigating more acute medication adjustments in polypharmacy patients.

### CONCLUSION

An optimal approach for targeting patients with problematic polypharmacy is yet to be determined. To address the challenges posed by confounding, it may be valuable to develop a cross-cutting measure of polypharmacy that

consistently accounts for multimorbidity. The complexities of prescribing decisions in polypharmacy highlight the importance of improved approaches that consider patient perspectives, individual factors and clinical appropriateness.

**Author affiliations**
[1]Centre for Primary Care and Health Services Research, School of Health Sciences, The University of Manchester Division of Population Health Health Services Research and Primary Care, Manchester, UK
[2]NIHR Greater Manchester Patient Safety Research Collaboration (GMPSRC), Faculty of Biology, Medicine and Health, Manchester Academic Health Sciences Centre (MAHSC), The University of Manchester, Manchester, UK
[3]Division of Informatics, Imaging and Data Sciences, School of Health Sciences, The University of Manchester, Manchester, UK
[4]Department of Health and Community Sciences, University of Exeter Medical School, Exeter, UK
[5]Division of Pharmacy and Optometry, School of Health Sciences, Faculty of Biology Medicine and Health, The University of Manchester, Manchester, UK

**Contributors** All authors were involved in the conceptualisation and design of the study. JYT performed the search, selection, extraction and synthesis with contributions from TB. JYT wrote the first draft of the manuscript. All authors contributed to the content and revisions of the review and approval of the final manuscript. JYT is the guarantor for the study.

**Funding** JYT is funded by the NIHR Doctoral Fellowship Programme (Ref: NIHR302624) for this research project. TB and DMA are supported by the NIHR Greater Manchester Patient Safety Research Collaboration. The views expressed in this document are those of the authors and not necessarily those of the NIHR, NHS or the UK Department of Health and Social Care.

**Competing interests** DA reports research funding from Abbvie, Almirall, Celgene, Eli Lilly, Novartis, UCB and the Leo Foundation outside the submitted work. The other authors have no conflicts of interest to disclose.

**Patient and public involvement** Patients and/or the public were involved in the design, or conduct, or reporting, or dissemination plans of this research. Refer to the Methods section for further details.

**Patient consent for publication** Not applicable.

**Provenance and peer review** Not commissioned; externally peer reviewed.

**Data availability statement** All data relevant to the study are included in the article or uploaded as supplementary information.

**ORCID iDs**
Jung Yin Tsang http://orcid.org/0000-0002-0331-2777
Matthew Sperrin http://orcid.org/0000-0002-5351-9960

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
