## [Reviewer comments · BMJ Open]

ARTICLE DETAILS

TITLE (PROVISIONAL)	Defining, identifying, and addressing problematic polypharmacy within multimorbidity in primary care: A scoping review
AUTHORS	Tsang, Jung Yin; Sperrin, Matthew; Blakeman, Thomas; Payne, Rupert A; Ashcroft, Darren

VERSION 1 – REVIEW

REVIEWER	Sivanandy, Palanisamy International Medical University
REVIEW RETURNED	26-Nov-2023

GENERAL COMMENTS	Most of the references.
-------------------------

REVIEWER	Barry, Heather Queens University of Belfast, School of Pharmacy
REVIEW RETURNED	05-Dec-2023

GENERAL COMMENTS	Thank you for asking me to review this manuscript, which details a scoping review undertaken to explore and conceptualise how patients with problematic polypharmacy are targeted for intervention. The manuscript is comprehensive and written to a high standard. The aim and objectives are clear. The methods are appropriate and well described. My main issue is that the searches are now nearly a year out of date (January 2023) and it would be advisable to update these prior to publication. It was evident that a number of key references, which have been updated over the last several months, are out of date. Some additional, minor points: 1. Page 6, Line 130: Explain CINAHL acronym2. Page 6, Line 132: How were the search terms developed?3. Page 7, Line 136: You refer to looking for grey literature and then list three additional databases; I assume you were also looking for clinical trial records?4. Page 9, Line 161: Should read 'The data were extracted...'5. Page 9, Lines 174-177: Avoid starting sentences with a numeral. Numbers under ten should be spelled out in full.6. Page 13, Line 243: Criteria should be criterion (singular)7. Table 4 and throughout: it became evident that old versions of STOPP/START and AGS Beers criteria papers are referred to here. These have both been replaced by updated versions this year.8. Page 16, Line 297: Grammar - 'there have been questions...'9. Page 19, Line 373: Grammar - the failure of the implementation of interventions is...'10: Page 20: The novelty of the findings needs to be emphasised within the discussion. Some of the initial findings regarding variation
--

	in definitions etc was already known (although this scoping review does bring together the evidence) but the 'so what'/what does the study add, needs to come through more strongly. 11. Pages 24 onwards: Some references, such as #12, have been updated this year.
--	---

VERSION 1 – AUTHOR RESPONSE

Reviewer: 1

Dr. Palanisamy Sivanandy, International Medical University

Comments to the Author:

Most of the references

Many thanks, our references have been updated and multiple references have been modified

Reviewer: 2

Dr. Heather Barry, Queens University of Belfast

Comments to the Author:

Thank you for asking me to review this manuscript, which details a scoping review undertaken to explore and conceptualise how patients with problematic polypharmacy are targeted for intervention. The manuscript is comprehensive and written to a high standard. The aim and objectives are clear. The methods are appropriate and well described. My main issue is that the searches are now nearly a year out of date (January 2023) and it would be advisable to update these prior to publication. It was evident that a number of key references, which have been updated over the last several months, are out of date.

Many thanks for your review, detailed comments and expertise. We have performed an updated search (completed in Feb 24), and this has added a further 19 articles, with updates to the Cochrane review (Cole et al. 2023), as well as multiple criteria e.g. STOPP/START, Beer's etc. which continue to develop at a swift pace. Updates have been made to the methods, results, discussion and the figure (PRISMA flowchart).

Some additional, minor points:

1. Page 6, Line 130: Explain CINAHL acronym

This has been clarified to read "Cumulative Index to Nursing and Allied Health Literature (CINAHL)".

2. Page 6, Line 132: How were the search terms developed?

This has now been clarified in the text "Search terms were developed after a preliminary search of articles covering our population, concept and context of interest".

3. Page 7, Line 136: You refer to looking for grey literature and then list three additional databases; I assume you were also looking for clinical trial records?

This has been updated to read "Three additional databases were then searched for grey literature and clinical trial records"

4. Page 9, Line 161: Should read 'The data were extracted...'

This has been changed as stated.

5. Page 9, Lines 174-177: Avoid starting sentences with a numeral. Numbers under ten should be spelled out in full.

Thanks, this has been changed as stated.

6. Page 13, Line 243: Criteria should be criterion (singular)

This has been changed as stated.

7. Table 4 and throughout: it became evident that old versions of STOPP/START and AGS Beers criteria papers are referred to here. These have both been replaced by updated versions this year. Updated versions of these criteria papers have been changed in Table 4 and throughout.

8. Page 16, Line 297: Grammar - 'there have been questions...'
This has been changed as stated.

9. Page 19, Line 373: Grammar - the failure of the implementation of interventions is...'
This has been changed as stated.

10: Page 20: The novelty of the findings needs to be emphasised within the discussion. Some of the initial findings regarding variation in definitions etc was already known (although this scoping review does bring together the evidence) but the 'so what'/what does the study add, needs to come through more strongly.

Many thanks – we have rewritten our discussion to try and emphasise these points.

11. Pages 24 onwards: Some references, such as #12, have been updated this year.

Many thanks – We have reviewed all the review references in the paper and have updated these to the best of our knowledge.

VERSION 2 – REVIEW

REVIEWER	Barry, Heather Queens University of Belfast, School of Pharmacy
REVIEW RETURNED	24-Mar-2024
GENERAL COMMENTS	Thank you for making the requested revisions to this manuscript and also for updating the searches, which I appreciate takes time to complete. I am happy with the changes that have been made to the manuscript and recommend that this should be accepted for publication.